# A Role for the Proteasome Alpha2 Subunit N-Tail in Substrate Processing

**DOI:** 10.3390/biom13030480

**Published:** 2023-03-05

**Authors:** Indrajit Sahu, Monika Bajorek, Xiaolin Tan, Madabhushi Srividya, Daria Krutauz, Noa Reis, Pawel A. Osmulski, Maria E. Gaczynska, Michael H. Glickman

**Affiliations:** 1Department of Cancer Biology, Dana-Farber Cancer Institute, Harvard Medical School, Boston, MA 02115, USA; 2Faculty of Biology, Technion—Israel Institute of Technology, Haifa 3525433, Israel; 3Department of Molecular Medicine, University of Texas Health Science Center at San Antonio, San Antonio, TX 78229, USA

**Keywords:** proteasome, ubiquitin, 20S gating, proteolysis, substrate translocation

## Abstract

The proteolytic active sites of the 26S proteasome are sequestered within the catalytic chamber of its 20S core particle (CP). Access to this chamber is through a narrow channel defined by the seven outer α subunits. In the resting state, the N-termini of neighboring α subunits form a gate blocking access to the channel. The attachment of the activators or regulatory particles rearranges the blocking α subunit N-termini facilitating the entry of substrates. By truncating or mutating each of the participating α N-termini, we report that whereas only a few N-termini are important for maintaining the closed gate, all seven N-termini participate in the open gate. Specifically, the open state is stabilized by a hydrogen bond between an invariant tyrosine (Y) in each subunit with a conserved aspartate (D) in its counterclockwise neighbor. The lone exception is the α1–α2 pair leaving a gap in the ring circumference. The third residue (X) of this YD(X) motif aligns with the open channel. Phenylalanine at this position in the α2 subunit comes in direct contact with the translocating substrate. Consequently, deletion of the α2 N-terminal tail attenuates proteolysis despite the appearance of an open gate state. In summary, the interlacing N-terminal YD(X) motifs regulate both the gating and translocation of the substrate.

## 1. Introduction

Careful removal of unwanted or unneeded proteins is necessary for cell survival. Eukaryotes have developed a complex yet controlled mechanism for precise proteolysis [1,2]. A multi-subunit protease, the proteasome, is the primary protease that carries out selective intracellular ATP-dependent protein degradation [3,4,5]. Regulation of this proteolytic pathway occurs at multiple levels, starting with the selection of substrates, recognition by the proteasome, and, once bound, the rate at which they are processed. In eukaryotes, proteasomes are typically found as a mixture of two main species: a 20S that consists only of a catalytic core and a 26S that is made up of the same 20S catalytic core attached to one or two regulatory particles (a.k.a. 19S or PA700) [6]. The 20S proteasome retains the basal ability found in archaeal proteasomes to proteolyze substrates with a significant disordered component [7]. By containing several ubiquitin-binding sites and six ATPase subunits (Rpts) in its regulatory particle, the eukaryotic 26S proteasome is selective for ubiquitinated proteins and can unfold globular domains; hence, its substrate repertoire differs from that of the 20S proteasome [1,3]. Indeed, in most cell lines under standard growth conditions, the majority of proteasomes are found as 26S holoenzymes [3,8,9].

The 20S core particle is a cylinder-like structure composed of four stacked heptameric rings with C2 symmetry (α_7_β_7_β_7_α_7_) engendering a sequestered proteolytic chamber [6]. The two outer α rings are each composed of seven structurally similar yet distinct subunits (α_1–7_), and likewise, each of the two inner β rings is formed from seven similar subunits (β_1–7_). Only three of these β subunits contain functional proteolytic active sites (β_1,2,5_), while the outer α rings define a gated channel leading into the inner proteolytic chamber (Appendix A). Substrates traverse this channel on their way to being hydrolyzed by the proteolytic β active sites [6]. In the 26S resting state, the gate through the 20S CP α ring is closed, and the ATPase channel of the 19S RP is misaligned. Once a ubiquitin-conjugate is bound, ATPase conformational changes lead to gate opening and a contiguous channel through the ATPase ring to the catalytic core [10,11,12,13]. In this manner, regulatory complexes such as the 19S RP participate in channel gating through the 26S holoenzyme and facilitate substrate entry [14,15,16,17,18,19,20].

A gated channel at the entrance to the 20S is a feature of eukaryotic proteasomes [21,22,23]. In the simple homomeric archaeal 20S proteasome, the identical N-termini of α-ring subunits orient in a radial symmetry and thus do not necessarily regulate the 20S gate [18,20,24,25,26,27,28,29,30]. In contrast, in the closed-state eukaryotic proteasome, the N-termini of the seven different α-subunits extend into the center of the α-ring as each assumes a unique conformation that is accurately matched to form a plug. The longest N-terminal tail of α3 runs across the diameter in the closed state, while the N-termini of other subunits curve upwards or downwards at the epicenter (Appendix A). Support for the pivotal role of α3 in gating was provided upon truncation of its N-terminal tail [23]. An atomic-level structure of these mutant proteasomes identified a pore at the center of the α ring, and biochemical studies showed that these “open gate” proteasomes hydrolyze both peptides and proteins faster than WT [23,31,32,33,34]. During the transition from the closed to the open gate configuration, all N termini rearrange, pull away from the center, and interact through an invariant proline with a conserved tyrosine in the first α helix (H0) of their neighbor [11]. A low occurrence of spontaneous rearrangement has been observed in free 20S CP, and the gate opening could be allosterically induced by signals from the active centers [29,35,36,37,38]. However, in the 26S proteasome, the open gate is induced and stabilized by the insertion of HbYX motifs at the C-termini of ATPase subunits into corresponding lysine-pockets on the outer α-surface of the 20S CP [18,20,39,40,41,42,43]. The open gate does not necessarily imply that the α subunit N-termini are merely disordered or randomly rearranged. X-ray crystallography of a yeast 20S in complex with an activator (PA26 from T. brucei) identified a rearrangement in a cluster of four highly conserved residues (YD-P-Y) near the N-terminus of each α subunit that stabilizes the open conformation [30]. Support for a role of the N-termini in defining a stable open conformation came from single-site substitutions of Tyr8 or Asp9 situated in the flexible N-terminal tail of the archaeal α subunit that resulted in slower proteolysis even in the presence of proteasome activators [30], or from substitutions in a cluster of aromatic residues of α2 and α3 subunits in yeast that resulted in resistance to stress conditions that increase the need for unfolded protein response [11].

In order to provide further details of the contribution of different α-subunit N-termini to the stabilization of either closed or open states, we compared the effects of mutating a conserved tyrosine residue in the N-terminal tail of each of the seven α subunits and found that they invariantly contribute to the open state. An open gate conformation is stabilized by hydrogen bonds between the tyrosine side chain of each α subunit and aspartate in the following α subunit. Though not conserved, the third residue in the sequence (YD(X)) of each subunit points directly toward the epicenter and lines the gate. Specifically, phenylalanine of α2 at this position engages the translocating substrate, as we reestablished from the published structures. We conclude that the N-terminal tail of α2 plays a unique role in facilitating substrates processing of 26S proteasome.

## 2. Materials and Methods

### 2.1. Yeast Expression Plasmid Constructs for WT and Mutant α Subunits

Standard PCR protocols were used to amplify the α1–α7 genes, together with the promoter, from yeast genomic DNA (strain BY4741 EUROSCARF), and the fragments were cloned into pUC19. The clones were sequenced on both strands and found to be free of mutations. Single-site mutation inside the YD(X) motif or the entire tail deletion in each α subunit gene was performed using PCR methods. The mutated DNA fragments were cloned into the pUC19 plasmids, and the resulting clone was sequenced. A set of plasmids designed to express WT and mutant versions of the α subunits in yeast was constructed.

### 2.2. Construction of Yeast Strains Expressing WT or Mutant α Subunits

Haploid cells (EUROSCARF Mat a strain) containing a given α gene deletion covered by a URA3-marked CEN plasmid expressing the WT gene were constructed. LEU2-marked CEN plasmids, carrying WT or mutant versions of a given α subunit gene, were introduced into the appropriate strain. Upon FOA-reversion [44], strains were obtained with only the LEU2-marked plasmids. WT or mutant α subunit genes expressed from these plasmids complemented the corresponding chromosomal deletion mutants. Strains with WT plasmids served as a control for the mutant strains.

### 2.3. Non-Denaturing PAGE and In-Gel Activity Assay

Protein samples were resolved by nondenaturing PAGE. A single resolving gel layer was prepared from the buffer of 0.18 M Tris-borate pH 8.3, 5 mM MgCl_2,_ 1 mM ATP, 1 mM DTT, and 4% acrylamide-bisacrylamide (at a ratio of 37.5:1) and polymerized with 0.1% TEMED and 0.1% ammonium persulfate. The running buffer was the same as the gel buffer but without acrylamide. Xylene cyanol was added to protein samples before loading them onto the gels. This Non-denaturing gel was run at 120 V until the Xylene cyanol reached the bottom of the gel (approximately 2 h). An in-gel peptidase activity assay [45,46] was performed after non-denaturing gel electrophoresis of 26S holoenzymes and 20S CPs. The gel was incubated in 10 mL of 0.1 mM Suc- LLVY-AMC in Buffer A (25 mM Tris pH 7.5, 10 mM MgCl_2_, 10% Glycerol, 1 mM ATP and 1 mM DTT) for 10 min. Proteasome bands were visualized upon exposure to UV light (360 nm) and photographed with a Polaroid camera. To characterize 20S CP as an independent complex, extracts were incubated with 0.5 M KCl without ATP for 30 min, which resulted in 26S disassembly. In order to activate 20S CP peptidase activity, SDS was added to a final concentration of 0.02% and incubated for 10 min.

### 2.4. Atomic Force Microscopy (AFM)

Imaging with atomic force microscopy (AFM) was performed as previously described [36]. In short, a purified proteasome sample was diluted with 5 mM Tris/HCl buffer pH 7.0 to obtain nanomolar concentrations of the protein. About 3 µL of the sample was deposited on a freshly cleaved muscovite mica surface. After about 5 min, the proteasomes, which attached electrostatically to the mica, were overlaid with 30 µL of the buffer used for dilution and mounted in the “wet chamber” of the NanoScope IIIa (Bruker Inc., Santa Barbara, CA, USA). Imaging was performed in a tapping mode in liquid using oxide-sharpened silicon nitride tips on cantilevers with a nominal spring constant of 0.32 N/m (Bruker Inc.). The resonant frequency of the tip was tuned to 9–10 kHz, with an amplitude of 200–500 mV and a setpoint ranging from 1.4 V to 1.8 V. Fields of 1 µm^2^ were scanned at the rates of 2–3 Hz, with trace and retrace images collected. The apparent resolution of images collected in the height mode was 512 × 512 pixels, which translated into single-pixel dimensions of about 2 nm^2^. The height-mode imaging data remain “raw” since the whole fields were subjected only to plane-fit and 1st-order flattening with the NanoScope IIIa (v. 5.12r3) or NanoScope Analysis (v. 1.7) Software (Bruker Corp., Billerica, MA, USA). For display purposes, images of single particles were zoomed in from the fields, their brightness and contrast were adjusted, and the occasional scan lines were removed with the NanoScope IIIa or NanoScope Analysis. The “open,” “intermediate,” and “closed” gate conformations for the single 20S proteasome particles in the top-view position were distinguished as previously described [47,48,49]. Briefly, the 6-pixel scan across the center of the α-ring was considered a representation of the proteasome “α-face” with the central gate. Numerical values for the heights of these pixels (“scan lines”) were collected with a practical vertical resolution of about 1 Å. When a plot of the scan line revealed a local maximum, the gate (and the respective particle) was classified as “closed.” Accordingly, the presence of a local minimum (a central dip) indicated the “open gate.” In turn, if a scan line presented a concave function without a local minimum, the gate was assigned an “intermediate” status [47,48,49]. To distinguish between distinct forms of the “closed” gate, we performed a roughness analysis on the eight-pixel α face scan-lines of randomly selected single images of core particles, fifteen each from WT, α3α7ΔN, and α6ΔN (NanoScope Analysis and SPIP/Scanning Probe Image Processor v. 6.013; Image Metrology, Lyngby, Denmark). The RMS (root mean square) values were considered a measure of the flatness of the gate area.

### 2.5. Purification of 20S Proteasome from Yeast and Activity Measurements

Yeasts were harvested from exponentially growing cultures, washed in water, suspended in ice-cold homogenization buffer, and broken in a microfluidizer (Microfluidics). Proteasomes were isolated from the yeast lysate by a set of differential centrifugations followed by ion exchange and gel filtration chromatography [36]. All procedures were carried out at 4 °C or on ice unless specified. Proteasome activity in each purification step and in chromatographic fractions was tested with SucLLVY-AMC, as described [36]. In short, the crude lysate after cell breaking was clarified by 20 min centrifugation at 10,000× *g* and then centrifuged for 1 h at 100,000× *g*. The resulting supernatant was centrifuged for 7 h at 100,000× *g* to pellet high molecular weight proteins. The resulting pellet was re-solubilized by homogenization (Dounce homogenizer) in column buffer (50 mM Tris-HCl, pH 7.0, 20% glycerol). The resulting high molecular weight protein fraction was treated with a mixture of 1 mg/mL RNase A and RNase mix (1% vol/vol each) for 1 h at 37 °C, followed by 10 min centrifugation at 15,000× *g*. The pellet containing contaminating RNA fragments was discarded. The supernatant was filtered using a 200 µm syringe or a spin filter and subjected to anion exchange chromatography (Q Sepharose; Sigma-Aldrich, Burlington, MA, USA) with a gradient of 0–0.5 M NaCl in column buffer used to elute the proteasome. The proteasome-containing fractions were pooled, and the buffer was exchanged for 50 mM sodium phosphate, pH 7.0, and 20% glycerol, using spin filters with 100 kDa molecular weight cut-off. The sample was then subjected to column chromatography on hydroxyapatite using a phosphate gradient. The resulting proteasome-containing fractions were gel filtrated on Superose 6. The pure 20S proteasomes (protein concentration 0.2–0.5 mg/mL) were aliquoted and stored at –20 °C in 50 mM Tris-HCl buffer, pH 7.0, with 20% glycerol. Once thawed, aliquots were stored on ice and used within 48 h.

The peptidase activity of the 20S proteasome was measured with commonly used fluorogenic peptide substrates (Bachem, Calbiochem): succinyl-LeuLeuValTyr-4-methylcoumarin-7-amide (SucLLVY-MCA, for measuring ChT-L activity), butoxycarbonyl-LeuArgArg-MCA (BocLRR-MCA, for T-L activity), and carbobenzoxy-LeuLeuGlu-β-naphthylamide (Z-LLE-βNA, for PGPH activity). The substrates were routinely used at the final concentration of 100 µM, except for K_m_ and V_max_ determination, when a range of at least 6 concentrations from 20 µM to 150 µM was used [50]. Stock solutions of the peptide substrates were prepared in dimethyl sulfoxide (DMSO). Enzyme activities were determined fluorometrically based on reaction rates calculated from reactions run for 60 min at 25 °C [36].

### 2.6. In Vivo Protein Degradation by Pulse-Chase Analysis

In vivo protein degradation was measured by the pulse-chase method using GCN4-βgalactosidase or Ub-Pro-βgalactosidase as a model substrate, expressed from URA3-marked plasmids. Yeast sub62 strains were cultured in a minimal medium containing 2% EtOH, 2% glycerol, or 2% raffinose for EUROSCARF strains. 1% galactose and required supplements overnight, diluted in 10 mL medium to O.D._600_ 0.2 in fresh YPD, and grown another 4 hrs. After harvesting and washing with -ura/-methionine medium, cells were resuspended in 0.3 mL of the same medium and transferred to a screw-capped tube containing 1 mCi [^35^S]-methionine (NEN “Express”). Labeling was performed by incubation for 10 min at 30 °C. After spinning down for 10 min, removing the supernatant, and resuspending the cell pellet in 2.5 mL of growth medium + 10 mM methionine/10 mM cysteine, aliquots were taken at different time points; 0, 5, 10, and 20 min for Gcn4-βgalactosidase and 0, 12, 24, and 48 min for Ub-Pro-βgalactosidase. Each aliquot of 600 µL was added to the tube on ice containing 102 µL of 1.85 M NaOH/7.4% β-ME. Each sample was left on ice for 10 min, added to 42 µL of 100% TCA, and left on ice for another 10 min. Samples were spun for 10 min at 4 °C, washed with ice-cold acetone, and spun again at 4 °C for 5 min. After drying in a speed-vac, and resuspending in 100 µL of 2.5% SDS, 5 mM EDTA, and 1 mM PMSF, samples were heated to 90 °C for 10 min and spun for 5 min at room temperature.

### 2.7. Immunoprecipitation of Residual Substrate for Quantification

The 0.1 mL lysate was diluted with 0.9 mL IP buffer (1% Triton X-100, 0.15 M NaCl, 5 mM EDTA, 50 mM HEPES pH 7.4) + anti-β-galactosidase, and the samples were gently agitated for 2 h at 4 °C. The immunoprecipitated protein was collected by adding 20 µL of a protein A/G-Sepharose (50% suspension in IP buffer) and further incubation for 2 hrs. The Sepharose beads were centrifuged and washed 3 times with IP buffer + 0.1% SDS. The proteins were released from Sepharose by boiling in 30 µL of 2 times concentrated Laemmli sample buffer, separated by SDS-PAGE (12%), and visualized by phosphoimager.

## 3. Results

### 3.1. N-Terminal YD(X) Motif of the 20S α Subunits Contribute to the Basal Peptidase Activity of Proteasomes

In the closed gate conformation, all seven N-termini of 20S CP α-subunits project in an extended secondary structure toward the center of the ring, forming a mesh of interactions (Appendix A). All seven N-terminal sequences are notable for a YD(X) motif, conserved between eukaryotes and found even in the archaeal sequence (Figure 1A and Appendix A). To test the contribution of this conserved element to the stabilization of the closed conformation, we replaced the invariant tyrosine with alanine in all subunits. The whole-cell extract from the mutant yeast strains was resolved by non-denaturing PAGE to visualize proteasome peptidase activity (Figure 1B). Proteasome composition and activity in the resulting whole-cell extract were similar across mutants and WT. No discernable activity of free 20S proteasomes was detected. To compare the activities of WT and mutant 20S particles, free-20S CP was generated by disassembling 26S proteasomes upon exposure to high ionic concertation with 500 mM KCl [51,52,53]. The released free 20S CP from α4Y/A mutant showed higher basal peptidase activities relative to WT or all other Y to A mutants, even without further stimulation by SDS (Figure 1B).

Although all α-subunits harbor a tyrosine at an equivalent position in their N-termini, only a mutation of tyrosine in the α4 subunit led to an increase in basal 20S peptidase activity (Figure 1B). A metadata analysis of the 20S structure (pdb: 1 ryp) for the molecular interactions of these N-terminal residues reveals that the tyrosine of the α4 YD(X) motif forms a hydrogen bond with the aspartate of α3 YD(X) (Figure 1C). Mutating the equivalent aspartate of the α3 N-terminal tail shows similar stimulation of 20S peptidase activity (Figure 1B), supporting the importance of the H-bond between the YD(X) motifs of these two neighboring subunits. The only other YD(X) motif pair that is implicated in a similar H-bonding is between Tyr of α6 and Asp of α7 (Figure 1D); however, mutating the α6 tyrosine of its YD(X) motif did not affect the free 20S peptidase activity. In addition, the N-terminus of α4 is notable for projecting downwards into the 20S substrate translocation channel, positioning below the gate (Figure 1C). The only other N-terminus that is similarly positioned is that of α2, observed in both yeast and human 20S CP (Appendix A). Unlike α4, available model structures do not implicate α2 in H-bonding with neighboring YD(X) in the closed state of the 20S CP. The intricate orientations and H-bond pairing of the α-subunit N-termini are conserved in the closed state of mammalian 20S particles (Appendix A).

To evaluate the potential contributions of α2 and α6 N-termini to basal peptidase activity, we truncated the N-termini up to the reverse turn (Figure 2A), similar to what has been performed earlier for α3 and α7 [9,23,33]. Based on peptidase activity, no detectable changes to 30S/26S proteasome activity were observed; however, there was a noticeable increase in basal peptidase activities of 20S in extracts from α2ΔN, α3ΔN, or the double mutant α3/α7ΔN compared to WT (Figure 2B). Upon dissociation, free 20S CP of α2ΔN, α3ΔN, or α3/α7ΔN exhibit higher specific peptidase activity than WT (Figure 2C). Nevertheless, the free 20S CP from α6ΔN and α7ΔN 20S also showed higher peptidase activity than WT complexes upon further activation by mild treatment with 0.02% SDS (Figure 2D). The observed higher 20S peptidase activity is a possible outcome of an open gate state [9,23,32,33].

### 3.2. AFM Points to Differential Contributions of α Subunit N-Termini to 20S Gate Dynamics

Atomic force microscopy (AFM) enables to visualization of hundreds of 20S particles and categorizes them based on the dynamic state of the α-ring surface. Typically, 20S CP can be found in a mixture of closed, open, and intermediate conformers [47,48,49,54]. These three categories are identified in a representative AFM field of purified 20S particles from yeast (Figure 3A). To correlate between the peptidase activity measured for 20S particles in Figure 1 and Figure 2 and α-ring gate dynamics, we performed AFM imaging on isolated 20S particles from WT and α2ΔN, α6ΔN, α3ΔN, α7ΔN, and the double mutant α3/α7ΔN. For each mutant, we analyzed between 360 and 810 single particles imaged from 6 to 13 scanned fields. Conformer partitioning of 20S particles from WT yeast was 75% (±4%) closed, 17% (±4%) intermediate, and 8% (±3%) open (Figure 3B,C), in concordance with the distribution of human 20S proteasomes [47,48,49,54]. No closed-gate particles were detected for the α3ΔN mutant; the overwhelming majority were open, albeit a well-detectable subpopulation (13% ± 4%) displayed an intermediate conformer. The 20S proteasomes from the α7ΔN mutant were also heavily affected, with mostly open particles (66% ± 4%), a sizable intermediate population (26% ± 5%), and a low number of closed-gate forms (7% ± 2%). Likewise, no properly formed closed forms were identified for α3/α7ΔN 20S proteasomes; however, the majority were classified as a new form that we refer to as pseudo-open (Figure 3B,C). These double mutant particles appeared flatter than WT closed particles. The observation was confirmed by roughness analysis (a measure of flatness) of the images of the gate area, where the pseudo-open double mutant particles were significantly distinct from the closed control (Appendix A). For comparison, the flatness of closed α6ΔN particles was neither significantly different from the control nor from the double mutant. Partitioning of conformers from α2ΔN and α4Y/A mutants were similar, with more than double the number of complexes with open gate (17–18%) compared to wild type, a relatively large subpopulation of intermediate forms (25–27%) and 56–57% of the particles were distinctly closed (Figure 3B,C). Deletion of the α6 N-terminal tail also led to the mild elevation of intermediate (23% ± 3%) and open (10% ± 3%) particles at the expense of closed (66% ± 4%) forms in α6ΔN mutant as compared to the wild type.

3D rendering of a typical plot of the gate area in closed conformation is markedly convex (Figure 3A). This is likely a result of the top-positioned N-terminal tails, most notably those from α1, α6, and α7, interacting with the AFM probe and repelling it just above the closed gate. Since in the closed position, the N-terminal fragment of α3 subunit traverses most of the α-ring surface and crosses the center of the gate, deletion results in severe destabilization of the closed gate. The resulting open and intermediate particles display elevated peptidase activity (Figure 2). The N-terminus of α7 points upwards from the α-ring surface, explaining the apparent open state assigned by the AFM probe of α7ΔN 20S proteasomes (Figure 3B). When both α3 and α7 tails are deleted in the double mutant, the top layer of the gate is destabilized strongly enough for the AFM probe to start detecting the second layer of the gate closure, consisting of downward-pointing tails of α2 and α4. Indeed, the flatness of gate area images was a tell-tale of non-standard interactions between the AFM probe and the gate parts, defined as the pseudo-open conformation. Interestingly, deleting tails of other subunits—α2ΔN, α6ΔN, and even the single substitution of the conserved YD(X) tyrosine of α4 in α4Y/A—led to a greater portion of open and intermediate 20S conformers relative to wild type. The enhanced peptidase activity measure for the α6 and α4 mutants may be correlated with the role of their N-termini in securing neighboring subunits in the closed state (Figure 1). Deletion of the α2 N-terminal tail also led to enhanced peptidase activity and seemingly greater gate dynamics, although it is not documented to form H-bonds with other tails. Therefore, we focused on the contribution of this tail to proteolysis.

### 3.3. A Role of the α2 N-Terminal Tail in Intra-Cellular Ubiquitin-Dependent Degradation

In the closed state, since the N-terminal tail of α2 is buried below the top layer of the gate, without contacting other N-termini (Figure 1C), we wished to understand whether the elevated peptidase activity displayed for the α2ΔN 20S CP (Figure 2) holds for larger substrates. To do so, we compared intracellular proteolysis rates in α2ΔN and WT strains by pulse-chase experiment using two different short-lived substrates (Figure 4A). Degradation of GCN4-βgal and Ub-Pro-βgal was slower in the α2ΔN mutant compared to WT (Figure 4B,C). Likewise, in α6ΔN, the degradation rates were slightly decreased compared to WT. In contrast, the α3/α7ΔN mutant degrades these two substrates faster than WT (Figure 4B) and [9,33,55,56], in accordance with its enhanced peptidase activity in Figure 2, supporting the critical role of α3 and α7 N-terminal tails in maintaining the closed gate conformation. PolyUb-conjugates accumulated in the α2ΔN mutant (Figure 4D), suggesting that the 26S proteasome is rate-limiting for the conjugate removal in this mutant strain. Inefficient proteolysis was further demonstrated by the sensitivity of α2ΔN and α6ΔN strains to the amino acid analog Canavanine (Figure 4E). The combined phenotypes of the α2ΔN mutant suggest that the N-terminal tail of α2 potentially regulates proteasome degradation function distinctly from other tails that participated in the closed gate state.

### 3.4. The Contribution of the N-Terminal YD(X) Motif to the 20S Open Gate Conformation

It was somewhat perplexing that the removal of the α2 N-terminal tail did not accelerate proteolysis, unlike a similar removal of the tail from α3. Following the binding of a ubiquitin-conjugate, the 26S proteasome goes through a series of conformational changes from a ground state to a substrate-engaging/-processing state that is accompanied by a 20S CP gate opening [3,11]. Examining the published substrate-engaged open gate state from the Martin or Mao groups [12,13] reveals that all seven α-subunit N-termini retract away from the center of the ring, pointing towards the ATPase channel (Figure 5A,B). The most dramatic movement is by the N-terminal tails of α2 and α4 that shift from a buried position in the second layer (Figure 1, Appendix A) to the top layer alongside all other N-termini (Figure 5A). Metadata analysis of pdb:6ef3 reveals that all seven N-termini are attached to one another through the H-bonding of their YD(X) motifs, forming a well-defined “N-ring.” The equivalent H-bonds are between Y and the D of the +1 neighbor (Figure 5B). To obtain this bonding interaction, Y of α4 switches its H-bonding from D of α3 in the closed state to D of α5 in the open state (and likewise, D of α3 forms a new bond with Y of α2). The result is that only in the open state each N-terminal tail is fixed by two H-bonds with both adjacent neighbors. The lone exception is the α1–α2 pair which does not interact in this manner due to the replacement of aspartate with serine in α2 (Appendix A).

Reorientation of the N-termini in the open state is characterized by the repositioning of all X residues (of the YD(X)) (Figure 5B). All seven residues are on the same plane, pointing inwards in an equivalent manner lining the central pore. Of note, four of the residues are positively charged or hydrophilic (five in human proteasomes), whereas two highly conserved residues are large hydrophobic amino acids (Phenylalanine in α2; Leucine in α7). To summarize, the open gate state is not merely a disruption of the N terminal tails positions in the closed state but a distinct arrangement of its own. In the open state, the N-termini are not “flexible” but rather locked into a given conformation (the N-ring) that may reflect their role in substrate translocation.

### 3.5. A hydrophobic Interaction of the α2 N-Terminal Tail with a Substrate

Thorough insight to substrate processing by the 26S proteasome was advanced by two recent studies that imaged a trapped substrate polypeptide in the translocation channel [12,13]. In the substrate processing state, the substrate backbone was traced from the ATPase pore to the 20S α-ring. The 20S gate is in an open state with all seven N-termini locked into the N-ring, forming a contagious channel with the ATPases (Figure 6A). Upon further analysis, interestingly, we observed in both human and yeast 26S structures, the substrate was found close to the α2 N-terminal tail. Specifically, in yeast proteasome, the substrate interacts with Phe7 of α2 (Figure 6B), and likewise, the translocating substrate contacts Phe8 of human α2 (Figure 6C). Both interactions are in the form of van der Waals interactions between hydrophobic groups (Appendix A). The nature of this interaction is made possible due to the invariant Phenylalanine at position “X” that lines the channel among all the YD(X) motifs (Appendix A). This interpretation supports our observation that the deletion of α2 N-terminal tail attenuated substrate degradation by 26S proteasomes (Figure 4), consistent with a role for α2 N-terminal tail in substrate translocation.

## 4. Discussion

This study identifies a motif in the N-termini of 20S α subunits that plays a dual role in both gating the translocation channel and facilitating substrate translocation. A prominent feature of eukaryotic 20S proteasomes is divergent and flexible N-termini that nevertheless share a conserved motif, referred to in this study as the YD(X) motif. These N-terminal tails inter-lace over more than one plane, hindering substrate entry in the 20S closed state yet pulling away from the center, defining an adequate pore for substrate entry in the open state. Rather than being merely disordered in the open state, this study provides evidence for a structured open state and participation of the N-terminal tails in regulating both the closed and open conformations of 20S CP. The concept of neighboring Y-D residues interacting in both closed and open states has been raised before [11,30]; here, we add that in the transition between closed and open states, some H-bond break, enabling the same residues to partner up with different neighbors. Specifically, interactions between three neighboring Y-D residues lock the gate in the closed state of the 20S translocation channel. Interactions between six Y-D pairs similarly stabilize the open gate conformation. The third residue of this motif-X, which lines the translocation channel, has a role in the 20S open state to interact with the translocating substrate.

Disordering of the N-terminal tails alone is insufficient to allow unobstructed entry through the pore, and the blocking residues may need to be removed or anchored into a stable, open conformation to relieve obstruction of the channel. An implication is that the open gate, as defined by structural methods such as AFM, does not necessarily correlate with biochemical properties such as proteolytic activity. This property is exemplified by deleting the N-terminal tail of α2, which appears partially “open” by AFM and is indeed enhanced for peptidase activity, yet in sharp contrast to α3ΔN proteasomes, displays slower ubiquitin-conjugate turnover in vivo. This suggests that conditions enhancing peptidase activity do not necessarily correlate with proteolytic efficiency due to the need for engagement of critical α-ring N-terminal tail residues during translocation of a long polypeptide.

Interestingly, in the closed state, α3 N-terminus does seem to serve as the linchpin. To a large extent, this could be attributed to its tight interaction with the N-terminal tail of α4. Rearrangement necessitates the breaking of some key interactions that anchor the tails in the closed conformation while forming competing new interactions to stabilize them in an open state. The most dramatic movement is of α2 and α4 N-termini that flip from being buried below the surface (Figure 1) to being above the surface as part of the α N-ring (Figure 5). In both cases, their YD residues establish new H-bonds. In the case of α4, this transition is also accompanied by breaking H-bonds with neighbors formed in the closed state. This could explain how single-site substitutions in α3 or α4 tails are sufficient to destabilize the closed state. The important conclusion is that the open state of 20S CP is well-defined through the interlocking of the α-subunit N termini, as can be easily visualized in the published structures, PDB:6ef3 or PDB:6msk (Figure 5 and Figure 6). Interestingly, our metadata analysis highlights that in the α N-ring, all nearest neighbors are locked except for α1–α2, which leaves room for flexibility and lateral movement in the ring. This may accommodate side chains or residual secondary structure of the substrate polypeptide or may be linked to their positioning in the translocation channel above the β2 active site in the proteolytic chamber, next layer. Additionally, in the open gate state, α2 interacts with the translocating substrate. The identity of X in the YD(X) motif defines the nature of interaction with substrates at the pore: for now, we know only of hydrophobic interactions between substrate and the invariant phenylalanine of α2. However, the roles of other X residues in the rest of the α subunits may become apparent with other substrates or other steps of the translocation.

## 5. Conclusions

The 20S proteasome is a self-compartmentalized protease that carefully restricts the free access of substrates and the continuous generation of products. The 20S α subunits stabilize both closed and open states by reorganizing specific interactions between neighboring residues in their N-terminal tails. A characteristic YD(X) motif in these tails is a hotspot for interaction switches between closed and open conformations. The N-tail of the α2 subunit predominantly participates in substrate proteolysis, most likely by directly interacting with the substrate. Through such interactions with substrate, the α subunit N ring defines not only substrate selectivity but possibly even cleavage specificity and the nature of products generated. By mutating the X residue of the YD(X) motif in alpha2 and in the other critical alpha subunits, one could map their relative responsibilities in substrate translocation and subsequent proteolysis.

## Figures and Tables

**Figure 1 biomolecules-13-00480-f001:**
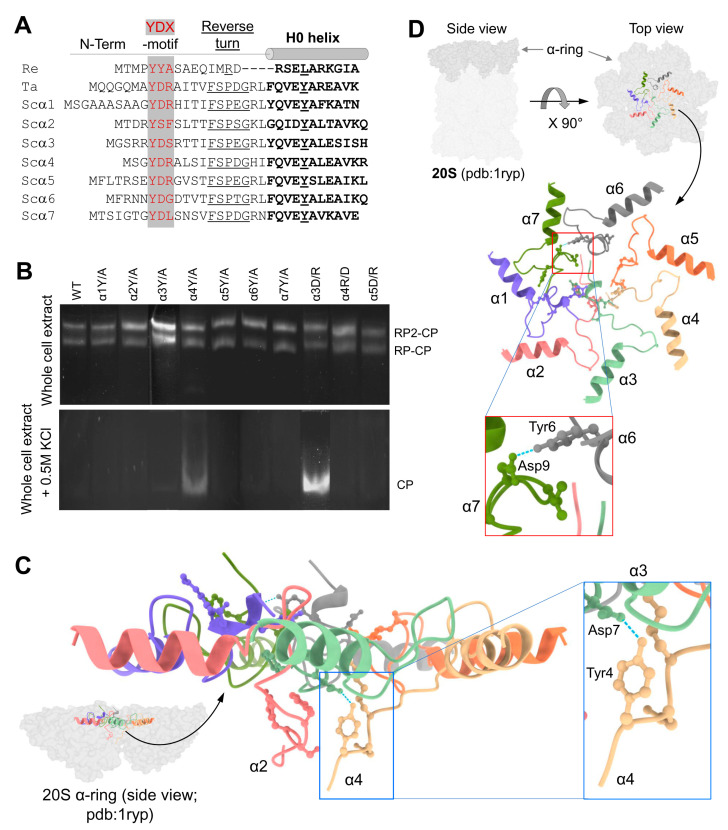
YDX motif of α-subunits is important for 20S closed-gate stability. (**A**) Sequence alignment of the N-terminal regions of 7 α-subunits of *Saccharomyces cerevisiae* (Sc) proteasome and the single α-subunits of *Thermoplasma acidophilum* (Ta) and *Rhodococcus erythropolis* (Re) proteasomes. The conserved YDX motif, Reverse Turn and H0 helix are marked. (**B**) Yeast mutant strains expressing 10 single point mutations within the YD(X) motif of respective α subunits were natively lysed, and peptidase activity of the 26S holoenzyme was monitored by non-denaturing gel electrophoresis (top panel). In each strain, tyrosine 8 of one α-subunit was substituted by alanine. Additionally, in some of the α subunits, also Aspartate 9 or Arginine 10 was substituted. Adding 0.5 M KCl and incubation of the cell extracts at 30 °C for 30 min resulted in 26S disassembly, and peptidase activity of the stable 20S CP was visualized (bottom panel). (**C**) Side view of α-ring (pdb:1ryp) highlights the lower plane of the α2 and α4 N-termini conformations in the closed state. The blue dash lines represent the H-bond between the α4Y and α3D residues of their YD(X) motifs. (**D**) A cartoon representation of resting 20S (pdb:1ryp) α-ring with surface view shows the N-terminal regions up to the H0-helix. Inserts show a zoom-in of the hydrogen bonds between the α6Y–Dα7 pair.

**Figure 2 biomolecules-13-00480-f002:**
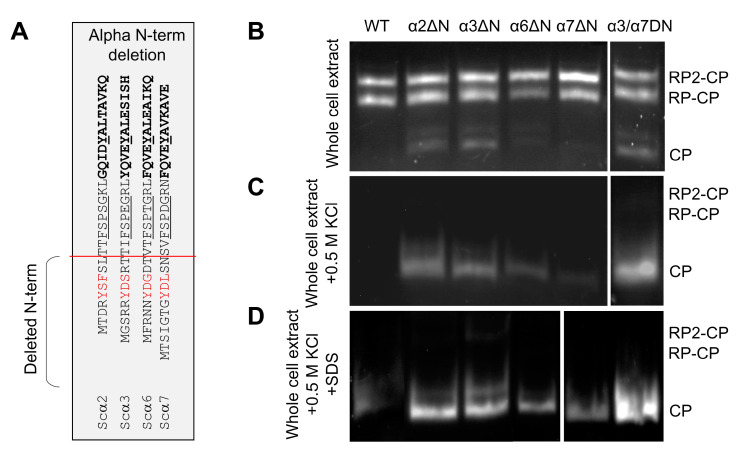
Effect of α-subunit N-terminal truncation on proteasome peptidase activity. (**A**) In the table, the red line shows the position of the amino acid sequences deleted from the N-termini of the α subunits to make the corresponding ΔN-mutant strains. (**B**–**D**) Non-denaturing gel electrophoresis of natively lysed whole cell extract followed by in-gel peptidase activity assay to visualize 26S holoenzymes and 20S CPs. Symmetric (RP_2_CP) and asymmetric (RP_1_CP) forms of the holoenzyme as well as the CP form, are indicated from the crude extract of WT and ΔN mutants (**B**) after adding 0.5 M KCl and incubation of the cell extracts in 30 °C for 30 min (**C**), and at the above condition in C supplemented with 0.02% SDS (**D**).

**Figure 3 biomolecules-13-00480-f003:**
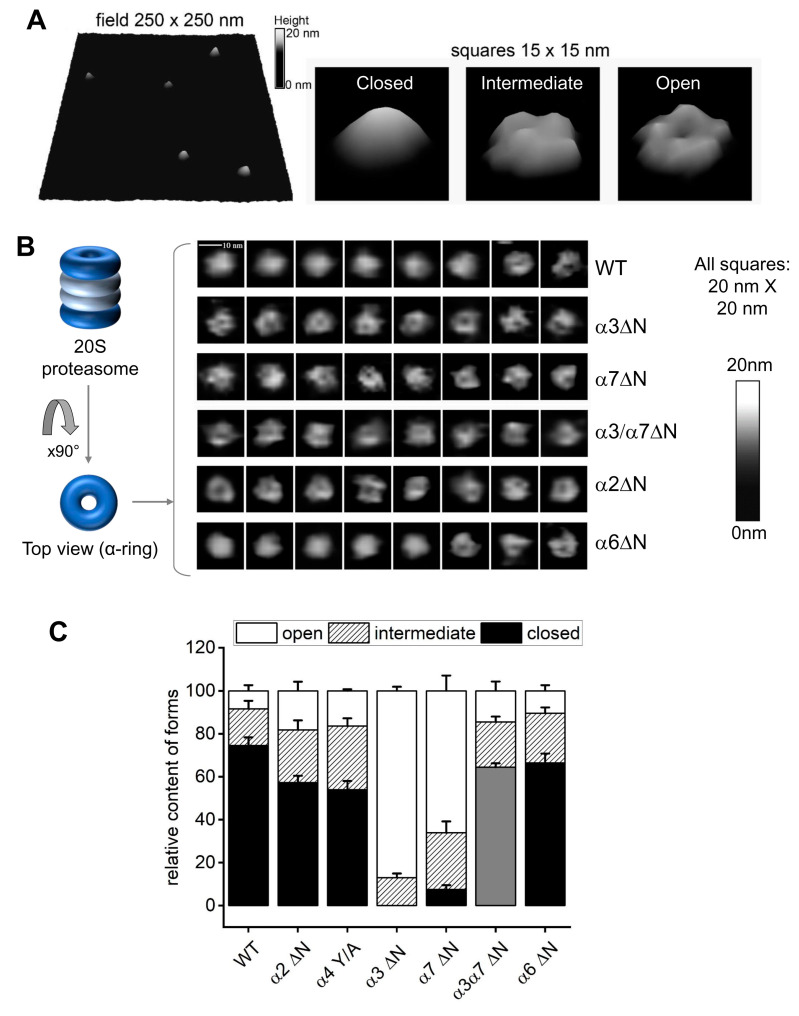
AFM images point to conformational diversity of the proteasome α-ring surface. (**A**) Pseudo-3D renderings of AFM images of top-view proteasome particles. Left: a fragment of the field with WT proteasomes. Right: representative images of closed, intermediate and open conformational forms. Shades of grey represent the height of imaged particles. Digital lateral resolution of images: 1.95 nm. (**B**) Atomic force microscopy (AFM) delivers images of the top surface of the α-ring (α-face) of core proteasome particles. Left: cartoon of the four-ring proteasome core particle inside view and top view positions. α-rings are colored dark blue and presented in open-gate conformation. Right: representative AFM images of 20S proteasome particles in top view positions. The majority of WT particles were imaged with a smooth, cone-shaped α ring, most likely representing the most stable closed-gate conformation. Such particles were not observed in preparations of the α3ΔN mutant. Instead, the majority of images from this mutant present a depression in the central part of their α ring constituting the gate area, presumably indicating an open gate. Such images were also prevalent among the α7ΔN mutant proteasomes. In turn, images of other mutant core particles presented diverse sets of cone-shaped, donut-shaped or irregular α-faces, with the α6ΔN mutant closely resembling the wild-type. Shades of grey represent the height of imaged particles. Digital lateral resolution of images: 1.95 nm. (**C**) The imaged proteasome particles were classified into three major conformational forms: closed, intermediate and open, based on computational analysis of sections through their α-faces [47,48,49]. Wild-type particles were imaged mostly in closed-gate conformation, with minor populations of intermediate and open-gate particles. The populations of intermediate and open-gate forms became more numerous in α2ΔN and α4Y/A mutants. Open-gate particles were prevalent among α3ΔN and α7ΔN mutant proteasomes. The grey-colored forms in α3α7ΔN represent “pseudo-open” particles with disturbed α-face. Mean + SD values are presented and computed for n fields with a total of x particles. From left (WT) to right (α6ΔN), the n-x values are 11–688, 8–446, 6–361, 13–625, 12–810, 8–546, and 8–443.

**Figure 4 biomolecules-13-00480-f004:**
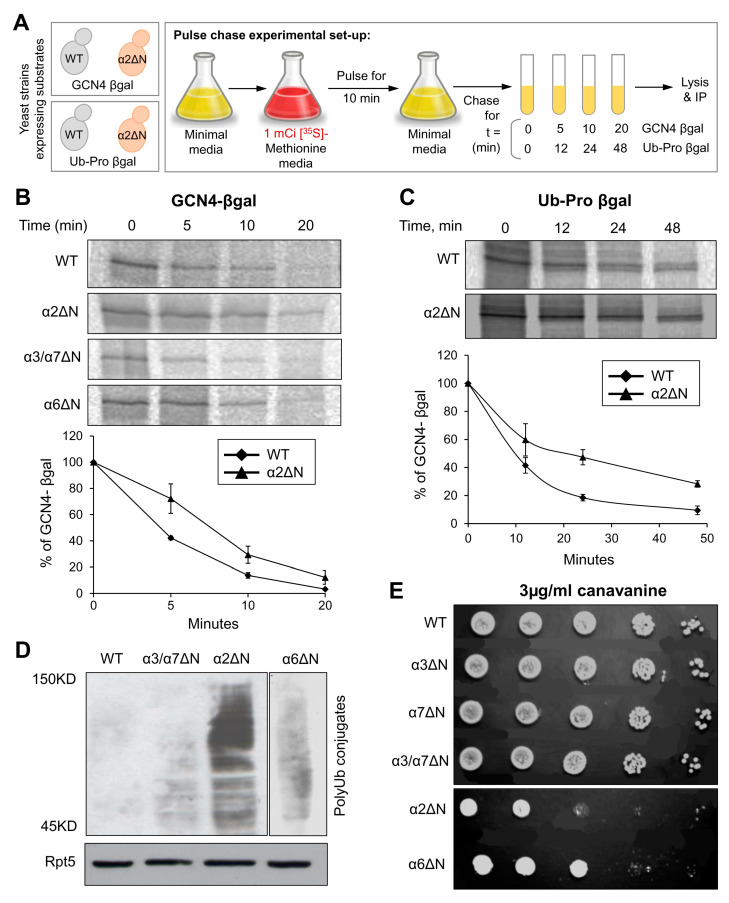
The α2ΔN mutant shows defects in intracellular proteolysis. (**A**) Pulse-chase experimental setup. GCN4-β-galactosidase or Pro-β-galactosidase substrates were expressed in WT and α subunit ΔN mutants. After a 10 min pulse with [^35^S]-methionine, excess unlabeled Met was added to initiate the chase incubation (t = 0). At the time indicated, cells were rapidly lysed, and the relevant substrate was immunoprecipitated. (**B**) GCN4-β-galactosidase from WT, α2ΔN, α6ΔN and α3/α7ΔN strains; and (**C**) Ub-Pro-β-galactosidase from WT and α2ΔN strains were immunoprecipitated by anti-β-galactosidase antibody and resolved by SDS-PAGE and quantified by a phosphoimager. Line graphs represent% of GCN4-β-galactosidase or Ub-Pro-β-galactosidase depletion estimated from three independent experiments each. (**D**) Ub-conjugates in WT and mutant strains indicated. Whole-cell extracts from different strains were separated by SDS-PAGE and immunoblotted with an anti-Ub antibody. Immunoblotting with anti-Rpt5 indicates identical proteasome levels in these extracts as a control. (**E**) Growth rates of WT and α subunit ΔN mutant strains during moderate stress of canavanine. Growth rates were tested by serial dilution onto 3 µg/mL canavanine plates.

**Figure 5 biomolecules-13-00480-f005:**
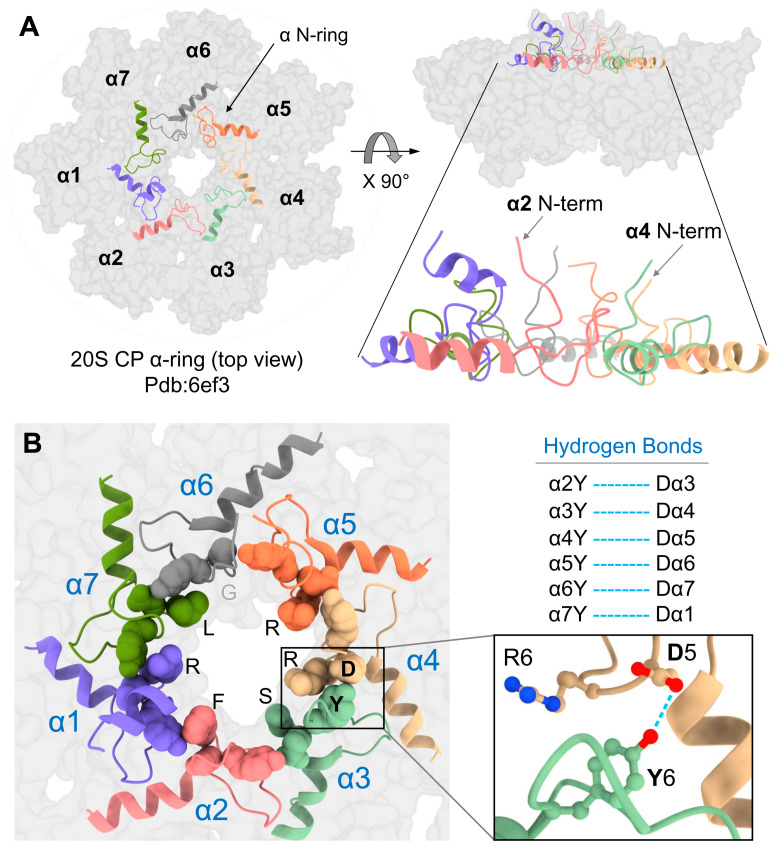
YD(X) motif contributes to the stability of 20S open gate conformation. (**A**) Left: Top view of 20S-CP α-ring of pdb:6ef3 structure showing in grey color surface contour. The N-terminal tails, including the H0-helix of all α-subunits (α N-ring), are shown in the colored cartoon display. Right: The side view of the same figure on the left highlights the upright-positioned N-terminal tails. (**B**) Left: a close-up view of the α N-ring showing the YD(X) motifs in sphere shapes and highlighting the X residues lining the inner chamber. Right: The corresponding H-bonds between the Y and D residues of all α-subunits (except for the α1–α2 pair, for which a corresponding H-bond was not identified). The blue dash lines represent the H-bonds between the Y and D residues of adjacent YD(X) motifs.

**Figure 6 biomolecules-13-00480-f006:**
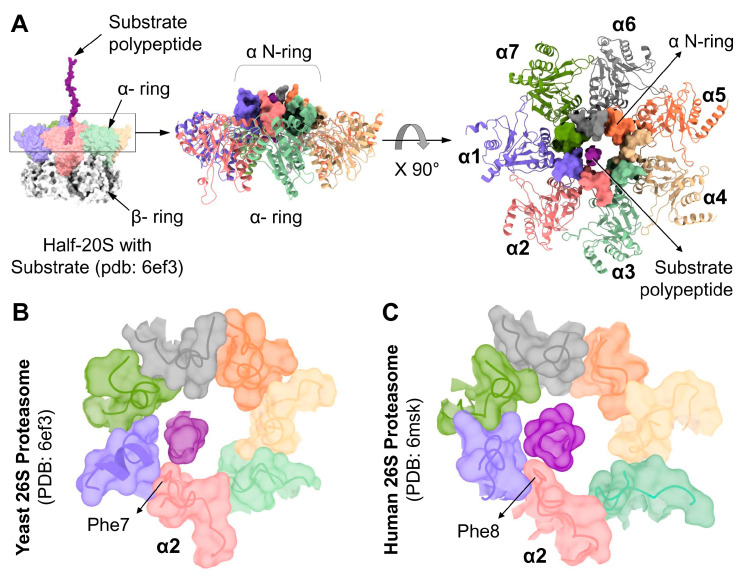
YD(X) motif of the α2 subunit interacts with the substrate during translocation into the 20S-CP. (**A**) Left: Surface contour structure of substrate-engaged yeast proteasome (pdb:6ef3) showing the substrate polypeptide entering the α N-ring. Right: Top view of the α N-ring in surface presentation while engaged with the substrate polypeptide. Top views of the α N-ring in surface presentation from 26S proteasome model structures: Yeast, pdb:6ef3 (**B**), and Human, pdb:6msk (**C**), showing the substrate polypeptide interacting with the Phe residues of the YDF sequences in the corresponding α2 subunits.

## Data Availability

The research data or other resources will be available upon request to the corresponding authors.

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
