# Peer review of "A Role for the Proteasome Alpha2 Subunit N-Tail in Substrate Processing"

_biomolecules, 2023, doi:10.3390/biom13030480_

Round 1
Reviewer 1 Report
In the manuscript entitled “A role for the proteasome alpha2 subunit N-tail in substrate 2 processing” the authors convincingly identify a YDX motif present at the N-terminus of proteasomal α-subunits that plays a crucial role in stabilizing the open conformation of the proteasomal gate and in facilitating translocation of substrates within the proteolytic cavity. Through well-designed and conducted mutagenesis, biochemical, atomic force microscopy and in vivo studies, Glickman and collaborators thoroughly investigate the molecular mechanism and the functional implications of this motif in regulating proteasomal structure and catalytic activities. This is undoubtedly an important study, the experimental approaches have been chosen appropriately and the conclusions are well supported by the experimental data. I strongly recommend the acceptance of the manuscript in its present form.
Minor remarks:
- Supplementary figures 1a and 1b are cited in several places in the text. In the version given to me, however, Supplementary Figure 1 does not have panels a and b. Please check and correct if necessary.
- Interestingly, and correlated with the results of the article, a role of residues Tyr8 and Asp9 in stabilizing the 20S open gate conformation (although through a similar but not identical molecular mechanism which also involves the residues Pro17 and Tyr26) had also been described based on the crystal structure of yeast 20S associated with PA26, the Trypanosoma homolog of mammalian PA28 (Forster A., et al. The EMBO J., 2003). I suggest to the authors to mention and briefly comment these data, because they indicate that Thy9 and Asp8 residues also play an important role in the activation of 20S induced by PA28s, which according to recent evidences could play an important role in the degradation of some classes of intracellular proteins.
Author Response
Reviewer 1
In the manuscript entitled “A role for the proteasome alpha2 subunit N-tail in substrate 2 processing” the authors convincingly identify a YDX motif present at the N-terminus of proteasomal α-subunits that plays a crucial role in stabilizing the open conformation of the proteasomal gate and in facilitating translocation of substrates within the proteolytic cavity. Through well-designed and conducted mutagenesis, biochemical, atomic force microscopy and in vivo studies, Glickman and collaborators thoroughly investigate the molecular mechanism and the functional implications of this motif in regulating proteasomal structure and catalytic activities. This is undoubtedly an important study, the experimental approaches have been chosen appropriately and the conclusions are well supported by the experimental data. I strongly recommend the acceptance of the manuscript in its present form.
Ans: We thank the reviewer for the optimistic comments.
Minor remarks:
- Supplementary figures 1a and 1b are cited in several places in the text. In the version given to me, however, Supplementary Figure 1 does not have panels a and b. Please check and correct if necessary.
Ans: The figure numbering is now corrected.
- Interestingly, and correlated with the results of the article, a role of residues Tyr8 and Asp9 in stabilizing the 20S open gate conformation (although through a similar but not identical molecular mechanism which also involves the residues Pro17 and Tyr26) had also been described based on the crystal structure of yeast 20S associated with PA26, the Trypanosoma homolog of mammalian PA28 (Forster A., et al. The EMBO J., 2003). I suggest to the authors to mention and briefly comment these data, because they indicate that Thy9 and Asp8 residues also play an important role in the activation of 20S induced by PA28s, which according to recent evidences could play an important role in the degradation of some classes of intracellular proteins.
Ans: Thank you for pointing this out. We expanded the attention given to this important study, and is now referenced both in the introduction and in the discussion.
Reviewer 2 Report
This manuscript presents a metaanalysis of previously published
structures of various 20S proteosome core particles combined with new
mutanagenesis data examined via biochemical assays and low-resolution
AFM to attempt to characterize important roles of the N-terminal tails
on the outer heptameric ring domains of these particles. The paper is
well-written and compelling, in interpreting the importance and
function of a tripeptide stretch on the 7 nonequivalent subunits.
The only direct structural information in the study comes from
AFM- a fairly low-resolution approach, so all of the atomistic details
come from the previously published crystal and cryo-EM structures. It
would be quite helpful if a concise summary of these data were given
in the introduction. Throughout the manuscript, clear distinction
between atomistic data, and interpretation should be given. This comes
to a head in the conclusion, with the statement (P. 15, line 492)
``The most important finding is a well-defined open state...''. This
result is presumably found in the structures 6ef3, 6msk, or is this region
not well resolved there. Whatever the case, this should be made clear.
Minor:
p. 3, line 139: ``pixel dimensions from nearly 2 nm^2'' : should this be
``pixel dimensions of about 2 nm^2'' ?
p. 9, line 367: ``found in closed'': should this : ``found close'', if
not it should be clarified.
Author Response
Reviewer 2
This manuscript presents a metaanalysis of previously published structures of various 20S proteosome core particles combined with new mutanagenesis data examined via biochemical assays and low-resolution AFM to attempt to characterize important roles of the N-terminal tails on the outer heptameric ring domains of these particles. The paper is well-written and compelling, in interpreting the importance and function of a tripeptide stretch on the 7 nonequivalent subunits.
The only direct structural information in the study comes from AFM- a fairly low-resolution approach, so all of the atomistic details come from the previously published crystal and cryo-EM structures. It would be quite helpful if a concise summary of these data were given in the introduction. Throughout the manuscript, clear distinction between atomistic data, and interpretation should be given. This comes to a head in the conclusion, with the statement (P. 15, line 492). The most important finding is a well-defined open state...''. This result is presumably found in the structures 6ef3, 6msk, or is this region not well resolved there. Whatever the case, this should be made clear.
Ans: We thank the reviewer for bringing this aspect. We have now made some relevant changes in the texts to highlight those aspects and the PDBs are clearly labeled in all figures, figure legends and necessary clarification is added in the text (for instances; Line 345 and line 495).
Minor:
p. 3, line 139: ``pixel dimensions from nearly 2 nm^2'' : should this be pixel dimensions of about 2 nm^2'' ?
Ans: Corrected as suggested.
9, line 367: ``found in closed'': should this : ``found close'', if not it should be clarified.
Ans: Corrected as suggested.
Reviewer 3 Report
The manuscript is worth addition in the field of substrate recognition and binding however there are some issues that must be addressed before the potential publication.
1. the authors failed to provide information regarding the 3D structures used in this study. please provide detail information.
2. the quality of Figure 1C and 1D seems very low. please improve the quality for better visualization and understanding.
3. in Figure 1C the numbering of the residues is important. Asp is not clear which Asp please provide the residue number too.
4. same problem with Figure 5.
5. separate conclusion should be added that contains information regarding the hot findings, gap, and future direction of the work.
Author Response
Reviewer 3
Comments and Suggestions for Authors
The manuscript is worth addition in the field of substrate recognition and binding however there are some issues that must be addressed before the potential publication.
- the authors failed to provide information regarding the 3D structures used in this study. please provide detail information.
Ans: We have now added in the text at relevant places the information on the published 3D structures those are used for metadata analysis in the current study. Reference to those PDBs are now mentioned in the text, Figures and Figure legends.
- the quality of Figure 1C and 1D seems very low. please improve the quality for better visualization and understanding.
Ans: We have replaced the corresponding figures with improved format for better visualization and understanding.
- in Figure 1C the numbering of the residues is important. Asp is not clear which Asp please provide the residue number too.
Ans: Corrected. Residue number added.
- same problem with Figure 5.
Ans: Corrected. Residue number added.
- separate conclusion should be added that contains information regarding the hot findings, gap, and future direction of the work.
Ans: Now necessary descriptions are added in the new section “5. Conclusion” in Page 17.
Round 2
Reviewer 3 Report
/